# Revealing the Importance of Aging, Environment, Size and Stabilization Mechanisms on the Stability of Metal Nanoparticles: A Case Study for Silver Nanoparticles in a Minimally Defined and Complex Undefined Bacterial Growth Medium

**DOI:** 10.3390/nano9121684

**Published:** 2019-11-25

**Authors:** Ilse De Leersnyder, Leen De Gelder, Isabel Van Driessche, Pieter Vermeir

**Affiliations:** 1Laboratory of Chemical Analysis (LCA), Department of Green Chemistry and Technology, Faculty of Bioscience Engineering, Ghent University, 9000 Ghent, Belgium; 2Department of Biotechnology, Laboratory for Environmental Biotechnology, Faculty of Bioscience Engineering, Ghent University, 9000 Ghent, Belgium; 3Department of Chemistry, Sol-Gel Center for Research on Inorganic Powders and Thin Film Synthesis (SCRiPTS), Faculty of Sciences, 9000 Ghent, Belgium

**Keywords:** AgNPs, test medium, size, stabilization mechanism, aging, aggregation

## Abstract

Although the production and stabilization of metal nanoparticles (MNPs) is well understood, the behavior of these MNPs (possible aggregation or disaggregation) when they are intentionally or unintentionally exposed to different environments is a factor that continues to be underrated or overlooked. A case study is performed to analyze the stability of silver nanoparticles (AgNPs)—one of the most frequently used MNPs with excellent antibacterial properties—within two bacterial growth media: a minimally defined medium (IDL) and an undefined complex medium (LB). Moreover, the effect of aging, size and stabilization mechanisms is considered. Results clearly indicate a strong aggregation when AgNPs are dispersed in IDL. Regarding LB, the 100 nm electrosterically stabilized AgNPs remain stable while all others aggregate. Moreover, a serious aging effect is observed for the 10 nm electrostatically stabilized AgNPs when added to LB: after aggregation a restabilization effect occurs over time. Generally, this study demonstrates that the aging, medium composition (environment), size and stabilization mechanism—rarely acknowledged as important factors in nanotoxicity studies—have a profound impact on the AgNPs stabilization and should gain more attention in scientific research.

## 1. Introduction

Lately, the world has seen an exponential rise in the applications of metal nanoparticles (MNPs), leading to an increasing interest of researchers from different scientific disciplines [1,2]. During the synthesis of MNPs, stabilizing agents are added to prevent the interaction of MNPs with one another. These stabilizing or capping agents typically stabilize the MNPs through the absorption or covalent attachment of organic compounds. The stabilization can be achieved either sterically, electrostatically or with both types combined: electrosterically. Due to the Brownian motion and Van der Waals attractive forces, particles can move to each other and aggregate. Macromolecules, like the polyvinylpyrrolidone (PVP) polymer can sterically stabilize MNPs by attaching to the surface and form a ‘brush-like’ layer. This layer reduces the degree of freedom when the particles approach each other, leading to an energetic unfavorable state whereby the particles repel and remain stabilized. Electrostatic repulsion is obtained by the presence of charged groups on the surface of the MNPs, that cause the formation of an electric double layer (EDL). When the charge of the repulsive electrostatic forces is strong enough to overcome attractive Van de Waals forces, the MNPs are stabilized. Sodium citrate (NaC) is an example of an electrostatic capping agent. A last option of stabilization is through electrosteric repulsion by the addition of a polyelectrolyte—like branched polyethyleneimine (BPEI). This stabilization mechanism is a combination of steric stabilizing polymers with charged groups [3,4,5,6,7].

MNP stability is an important issue to consider because aggregates can affect the reactive surface area and, hence, the reactivity, bioavailability and toxicity of MNPs [4,8]. Considering the MNP stability simply does not include the stability within their application during storage, rather, it also comprises the MNP stability when they move from the application to their intentional or unintentional ‘target’. The stability in these ‘target’ environments is critical and can even be life-saving. MNPs, for example, can translocate intentionally or unintentionally from the application into the human cardiovascular system through inhalation, digestion, injection or dermal contact. To ensure its safety and efficacy, it is crucial to consider what is happening when the MNPs transit through the alimentary tract or blood vein, for example, because MNP aggregation within the gastrointestinal fluids or blood can lead to reduced bioreactivity or capillary blockade [9,10,11,12,13,14,15]. The unintentional appearance of MNPs in environmental matrices, such as soil or aquatic sediments, also can lead to aggregation processes [3,8,16]. A case study for the stability of different types of silver nanoparticles (AgNPs) in two bacterial growth media will be elucidated in this paper.

AgNPs are one of the most frequently used nanomaterials, mainly due to their good antimicrobial activity resulting in a wide variety of promising applications in health, cleaning and the food industry [1,2,17,18]. The release of Ag^+^ from the AgNPs has an important role in their antimicrobial activity. Actually, Ag-ions often are regarded as the main bioactive species of AgNPs [3,19,20]. Many studies indicate that smaller particles are more effective, due to the easier cell uptake and increasing surface area which leads to a faster Ag^+^ release [21,22,23]. Besides the size, the antimicrobial efficacy of AgNPs depends on the shape. Nanoplates and triangular AgNPs are proven to be more toxic than spherical NPs due to the increased surface and the presence of more active facets [23,24,25]. The increasing use of AgNP products entails that, besides their desired beneficial effects, humans and the environment are exposed to AgNPs potential adverse consequences [3,16]. Consequently, an increasing number of publications investigate the effects of AgNPs. However, review papers already hinted to the overall low quality of nanotoxicity studies which often report conflicting results. The overlooked factor is often the interaction of the MNPs with the used medium [1,4,15,26,27,28], leading to possible significant changes in the NP properties. These complex mixtures with a certain pH, ionic strength and often the presence of organic matter can influence the stability and, thereby, alter the toxicity of AgNPs. When nanoparticles are destabilized, they will tend to form aggregates. Consequently, the surface area will be reduced and the Ag^+^ release will be lower [1,4,29,30,31,32,33].

Previously, we demonstrated the influence of several commonly used growth media components on the antimicrobial effect of Ag^+^ on *Bacillus subtilis* [20]. Now we focus on the stability of AgNPs in three media. Electrostatically (NaC), sterically (PVP) and electrosterically (BPEI) stabilized AgNPs of various sizes will be suspended in their solvent, an undefined complex and a minimally defined bacterial growth medium. The latter IDL medium specifically was developed such that components leading to the unavailability of Ag^+^, and a subsequent decrease of toxicity, were excluded [20]. The stability in these media is investigated during 24 h by means of three complementary analysis techniques: (1) measuring the UV-VIS absorption spectra as an addition to visual observations, (2) by transmission electron microscopy (TEM) and (3) dynamic light scattering (DLS).

## 2. Materials and Methods

### 2.1. Silver Nanosuspensions

Sodium citrate (NaC) stabilized AgNPs (10 nm and 100 nm), AgNPs stabilized with polyvinylpyrrolidone (PVP) (10 nm and 100 nm) and branched polyethyleneimine (BPEI) stabilized AgNPs (100 nm) were provided by Nanocomposix (San Diego, CA, USA) and belong to the NanoXact product line. The 50 nm BPEI stabilized AgNPs were provided by Sigma–Aldrich (St. Louis, MO, USA). The silver concentration of each suspension was 0.02 mg mL^−1^ and all AgNPs were spherical.

### 2.2. Preparation of Different Media

The composition of the used defined minimal medium, appointed as the IDL medium, was similar to an M9 medium [34] but certain M9 salts, which contain ions that form insoluble products with Ag^+^, were replaced by other salts. Consequently, components leading to the unavailability of Ag^+^ and the subsequent decrease of toxicity were excluded in the IDL medium. Detailed information regarding the different media can be found in De Leersnyder et al. [20]. Concerning the preparation of the IDL medium, a concentrated salt solution was prepared by dissolving 15 g L^−1^ KH_2_PO_4_ (Sigma–Aldrich), 85.49 g L^−1^ Na_2_HPO_4_·2H_2_O (Merck, Darmstadt, Germany), 3.04 g L^−1^ Na_2_SO_4_ (UCB, Brussels, Belgium), and 6.18 g L^−1^ (NH_4_)_2_SO_4_ (Merck) in an autoclaved Milli Q^®^ (Merck). This concentrated IDL salt solution was 5 times diluted and MgSO_4_·7H_2_O (Sigma–Aldrich) and Ca(NO_3_)_2_·4H_2_O (Acros Organics, Waltham, MA, USA) were added as micronutrients to a final concentration of 2 mM and 0.1 mM, respectively. Glucose (Merck) was added as the carbon source in 0.4% w/v to the final volume. The undefined complex Luria-Bertani (LB) medium was prepared by dissolving 10 g tryptone (VWR, West Chester, PA, USA), 5 g yeast extract (VWR), and 10 g NaCl (VWR) in 1 L of Milli Q^®^. The 2 mM NaC solution was prepared by dissolving NaC (Merck) into Milli Q^®^.

### 2.3. pH and EC Measurement

The pH and electrical conductivity (EC) of both media was measured by the inoLab^®^ WTW pH Level 1 pH-meter (Xylem, New York City, NY, USA) with a SenTix^®^ 81 WTW electrode (Xylem) and the K611 EC-meter (Consort, Turnhout, Belgium) with a SK20T electrode (Consort), respectively.

### 2.4. Evaluation of the Stability

The different types of AgNPs were diluted in a 1:1 ratio. This dilution was made in their solvent—2 mM NaC solution for the NaC stabilized AgNPs and ultra-pure water (Milli Q^®^, Merck) for the PVP and BPEI stabilized AgNPs—, in an IDL medium or in an LB medium. The influence of these different media on the stability of the different AgNPs was investigated by means of three complementary analysis techniques: (1) measuring the UV-VIS absorption spectra as an addition to visual observations; (2) by TEM; and (3) DLS.

#### 2.4.1. UV-VIS Spectroscopy

UV-VIS spectroscopy was performed by a Genesys UV-VIS spectrophotometer (Thermo Scientific, Waltham, MA, USA). The absorption spectrum was recorded from 340 nm to 700 nm. The samples were measured during different time intervals between 0 and 6 h and after 24 h of incubation. The mixtures were stored at room temperature in quartz cuvettes and in day–night light conditions in between measurements. Also, blanks (without AgNPs) of an IDL and an LB medium mixed in a 1:1 ratio with the solvent specific for the stabilization type of the AgNPs were measured. Moreover, a visual analysis of the samples was performed in parallel.

#### 2.4.2. Transmission Electron Microscopy (TEM)

TEM was performed at 60 kV with the TEM JEM1010 (Jeol, Tokyo, Japan). Pictures were digitalized using a Ditabis (Pforzheim, Germany) system. As previously described, the AgNPs were diluted in a 1:1 ratio with the medium prior to TEM analysis. Two microlitres of sample was put onto the carbon coated TEM grid of 200 Mesh (Electron Microscopy Sciences, Hatfield, PA, USA) and air dried. TEM analysis was performed immediately after mixing the AgNPs and medium or after 24 h of incubation of this mixture in a 1.5 mL Eppendorf tube in day–night light conditions.

#### 2.4.3. Dynamic Light Scattering (DLS)

DLS was measured by the Zetasizer Nano ZS (Malvern Instruments, Malvern, United Kingdom) with Zetasizer software 7.11. After mixing the AgNPs with the medium—as previously described—the samples were analyzed by DLS after different time intervals between 0 and 6 h and after 24 h of incubation. Analogous to the UV-VIS spectrophotometry and TEM samples, the mixtures were stored at room temperature in PMMA cuvettes and in day–night light conditions in between measurements. Using the DLS software, silver was selected as the dispersed material. A refraction index (RI) of 0.150 and absorption of 0.001 was entered. Water was selected as the dispersant for all samples. Water has a viscosity of 0.8872 cP and RI of 1.330, at a temperature of 25 °C. Three measurements for each of the three runs were performed on each sample per time point.

## 3. Results

### 3.1. pH and EC

The pH and EC of both media was measured immediately after preparation. A pH of 7.06 ± 0.06 and 7.02 ± 0.04 was measured for the IDL and LB medium, respectively. Consequently, the pH of both media can be considered as equal and neutral. The LB medium had an EC of 19.69 ± 0.35 mS cm^−1^, while the IDL medium had a lower EC of 9.53 ± 0.28 mS cm^−1^. The high amount of NaCl in the LB medium, compared with the minimal medium, resulted in an increase in EC [20].

### 3.2. Evaluation of the Stability by UV-VIS Spectroscopy

AgNP suspensions show unique optical UV-VIS absorption spectra and have typical vibrant colors because they have free electrons in the conductivity band. Specific wavelengths of light can drive the conduction electrons in the metal to collectively oscillate, a phenomenon known as surface plasmon resonance. These vibrations are specified by the size and shape of the AgNPs. Therefore, UV-VIS spectroscopy can be used as a characterization technique that provides information about the size and shape of the AgNPs [35,36,37,38]. Small Ag nanospheres (10–50 nm) typically have a small absorbance peak near 400 nm, while larger spheres (100 nm) give a broader peak with a maximum that shifts toward longer wavelengths near 500 nm. Moreover, the spectra of larger spheres have a secondary peak at shorter wavelengths, which is a result of quadrupole resonance in addition to the primary dipole resonance [39,40,41,42]. Destabilization and the formation of aggregates can lead to peak broadening or a secondary peak will form at longer wavelengths. Actually, an ever-increasing aggregation finally will lead to the disappearance of the typical UV-VIS absorption spectra. Therefore, absorbance spectra give information as to whether the AgNPs suspension has destabilized over time. This change in absorbance spectra often will be visible as a color change [37,39,43].

The original color of the LB medium is pale yellow and IDL is colorless. Figure 1 shows the UV-VIS absorption spectra of the blanks (without AgNPs). Regarding all 4 mixtures, the spectra and color remained constant through time. Concerning the spectra of the mixtures with LB, a typical increase in absorbance at shorter wavelengths was observed. The spectra of mixtures with IDL gave a negligible signal for all tested wavelengths.

Considering all tested capping agents, the AgNPs of 10 nm or 50 nm had a bright yellow color, while the large AgNPs of 100 nm were cloudier and had a white-gray color. This original, less vibrant color led to an indistinct color change when aggregation occurred. Color change will, therefore, only be shown for the AgNPs of 10 and 50 nm.

Figure 2, Figure 3 and Figure 4 show the spectra of the NaC, PVP and BPEI stabilized AgNPs, respectively. When the particles were mixed solely with their solvent, the spectra remained constant through time, which indicates a stable suspension. The Ag nanospheres of 10 and 50 nm had a bright yellow color and a small absorbance peak with a maximum near 400 nm for the 10 nm and at 420 nm for the 50 nm AgNPs. The larger spheres of 100 nm gave a broader peak with a maximum that shifted toward 480 nm. Moreover, a secondary peak was observed in this spectrum at 400 nm. These typical AgNPs’ absorbance spectra were consistent with the literature [39,40,41,42]. Regarding some, a small decrease in absorbance in a function of time was observed when suspended in their solvent. The nanoparticles may settle during storage, leading to an obscure color change to a lighter color and a small decrease in absorbance of the whole spectrum.

Adding the 10 nm NaC AgNPs (Figure 2) to the IDL and LB media resulted in a sudden change in color: bright pink for the IDL medium and bright orange for LB. The color intensity decreased over time and, after 24 h, the color returned to colorless for IDL and to a light yellow for LB. Regarding the minimal IDL medium, the original 10 nm absorbance peak was noticeable, but another peak at a longer wavelength was formed instantly after mixing. After a slight shift to the right, a decrease in absorbance occurred across the whole spectrum as a function of time, indicating an aggregation process. Concerning the LB medium, the original absorbance peak of the 10 nm AgNPs was vanished almost completely by onset of the measurements, but the peak reappeared with time, which indicates a disaggregation process. The 100 nm NaC stabilized particles gave a slight decrease in cloudiness as time advanced and, after 24 h, the mixtures returned to the original IDL and LB colors. The original 100 nm spectrum still was noticeable slightly in both IDL and LB at the start but disappeared quickly in time.

Concerning the 10 nm PVP AgNPs (Figure 3), a pink color was observed in the IDL medium and a bright yellow color in the LB medium. Through time, the color intensity decreased to less intense pink for the IDL medium and a more grayish yellow for the LB medium. Considering the minimal medium, the original 10 nm AgNPs peak still was noticeable at first, but a plateau at a longer wavelength was formed instantly. When time advanced, a slight shift to the right occurred and the absorbance of the whole spectrum decreased, thus aggregation occurred. Regarding the LB medium, a similar result was observed: the original 10 nm AgNPs peak still was clearly noticeable at first but disappeared with time. As mentioned before, the color change of 100 nm was obscure: the intensity of the cloudiness decreased as time advanced and, finally, the AgNPs color disappeared. Concerning both the IDL and LB media, the original 100 nm spectrum was clearly visible during the first timepoints but, afterward, the whole spectrum decreased quickly in absorbance.

The original yellow color of the 50 nm BPEI AgNPs (Figure 4) changed immediately after mixing with IDL or LB: first a grayish color was observed in both media, followed by the return to the original medium color. The first measured spectrum showed—for both media—a spectrum where the original 50 nm AgNPs peak still was apparent at 420 nm, followed by a plateau at a longer wavelength. When time advanced the absorbance of the whole spectrum decreased and, thus, aggregation occurred. The larger BPEI stabilized AgNPs of 100 nm showed a slight decrease in cloudiness and, finally, a complete disappearance of the color in the IDL medium. Considering the LB medium, the cloudiness and color of the 100 nm AgNPs still was observed after 24 h of incubation. Regarding the absorbance spectra, the original 100 nm peak still was visible after mixing with the IDL in the beginning of the measurements, but sharply decreased when time advanced. After mixing the 100 nm BPEI AgNPs with the LB medium, the 100 nm AgNPs peak was visible and the spectrum stayed constant through time.

### 3.3. Evaluation of the Stability by TEM

After mixing the NaC, PVP and BPEI stabilized AgNPs with their solvent, TEM analysis was performed immediately and after 24 h (Figure 5, Figure 6 and Figure 7). No aggregation was observed for both time intervals. Furthermore, the AgNPs showed a spherical shape and their size was in accordance with the expectations. Additionally, TEM analysis showed that the LB medium gave some more background-like crystal structures compared with Milli Q, 2 mM NaC or IDL.

Images of 10 nm NaC AgNPs (Figure 5) in IDL display that aggregation occurred immediately after mixing and an increase in size of these aggregates was observed when time advanced. Similar observations were seen for the 100 nm NaC AgNPs mixed with IDL and LB. Conversely, 10 nm NaC AgNPs in LB initially resulted in aggregates but, after 24 h, the aggregates became smaller, indicating a disaggregation process.

Increasing aggregation also was observed for 10 nm and 100 nm PVP AgNPs (Figure 6) when suspended in IDL. Regarding the LB medium, results were different. Both sizes of the sterically stabilized AgNPs had single particles observed instantly after addition to the medium, and aggregated particles were visible after 24 h. Moreover, it was remarkable that the aggregates of the 10 nm PVP AgNPs had a more spherical shape compared to the other observed aggregates.

Aggregates were formed when the 50 nm BPEI AgNPs (Figure 7) were mixed with IDL and LB. Similar to the previous observations, the size of these aggregates became larger when time passed. Concerning the larger BPEI AgNPs, increasing aggregation was seen in IDL, but single particles were observed when the AgNPs were suspended in LB. Even after 24 h, the 100 nm BPEI AgNPs were not aggregated in this complex medium, indicating a stable nanosuspension.

### 3.4. Evaluation of the Stability by DLS

During the final stage, the UV-VIS spectroscopy and TEM measurements were substantiated through DLS analysis. Size distribution graphs and the polydispersity index (PdI) of each measurement are reported in this paper. The PdI is a value that ranges from 0–1. It is used to describe the width of the particle size distribution and gives information about the polydispersity of the sample. A PdI value that is higher than 0.400 indicates a polydisperse system. This means the sample may not be suitable for a DLS measurement and that the provided data may be unreliable [44,45]. Reporting of the hydrodynamic diameter of an aggregate alone seems incorrect to us due to the complexity of aggregation processes that can lead to more polydispersity and the formation of non-spherical aggregates of which size cannot be defined by one value. Through the combining of both PdIs and size distribution graphs, DLS can be a good addition to our previous results.

Figure 8, Figure 9 and Figure 10 show the size distribution graphs which represent the number percent in function of size (nm) of NaC, PVP and BPEI AgNPs, respectively. The PdI values of each measurement are listed in Table 1. Regarding all tested AgNPs, no remarkable shift in the size distribution graph was observed when the particles were suspended in their solvent. Moreover, the PdI values of these samples were smaller than 0.400, apart from one single measurement. Similar to the UV-VIS spectrophotometer and the TEM observation, DLS data confirmed that the used AgNPs remained stable during 24 h in their solvent.

Regarding the electrostatically stabilized AgNPs, PdI (Table 1) was increasing sharply for both tested sizes when suspended in the minimal IDL medium. When the 100 nm particles were mixed with LB, a similar pattern was observed. This indicates an increasing polydispersity, due to aggregation, leading to less reliable DLS results [44,45]. The size distribution graphs (Figure 8) shifted back and forth for these samples, at least partly due to the higher polydispersity. The first measurement, whereby the PdI was still below 0.400, showed a distribution around 1000 nm for the 10 nm NaC AgNPs in IDL, while the size distribution of the 100 nm NaC particles initially stayed situated around 100 nm in both IDL and LB. The PdIs of 10 nm NaC AgNPs mixed with LB were within the limit of 0.400, and the size distribution graph shifted from a larger to a smaller size, indicating a disaggregation process.

Analogous to the NaC AgNPs, PdIs of 10 nm and 100 nm PVP (Table 1) in IDL and 100 nm PVP in LB was rising above 0.400. Only the first measurements showed a PdI value below 0.400. The size distribution (Figure 9) of the 100 nm particles still was situated around 100 nm immediately after mixing with both media. Concerning the 10 nm AgNPs in IDL, a larger size was measured at the first timepoint. Considering these three samples, the profile shifts to the right, indicating an aggregation process. Ten nanometers PVP AgNPs in LB gave PdIs that stayed below 0.400 when time advanced, apart from the last measurement. The distribution curves showed that size was getting bigger.

Lastly, BPEI stabilized particles were analyzed by DLS. PdIs (Table 1) were increasing above 0.400 for the mixture of 50 nm and 100 nm AgNPs in IDL and the 50 nm in LB, with exception of the first timepoint. Regarding the size distribution graphs (Figure 10), an increasing size was measured, starting from the first measurement. No change in size distribution was observed when 100 nm BPEI AgNPs were suspended in LB. Furthermore, the PdIs of this sample stayed below 0.400 for the performed measurements, which indicate that the 100 nm BPEI AgNPs remained stable in LB.

## 4. Discussion

All three analysis techniques corroborated that the tested AgNPs remained stable during 24 h in their solvent. No change in absorbance spectrum or color was observed. Microscopy images showed no aggregates, and DLS gave acceptable PdIs and narrow size distributions curves without shifting in time. This control confirms that the performed storage conditions (light and temperature) had no effect on the aggregation state of the AgNPs.

When the electrostatically (NaC), sterically (PVP) and electrosterically (BPEI) stabilized AgNPs were mixed with the minimally defined IDL medium, aggregation occurred almost immediately. A remarkable color change occurred for the small AgNPs and all spectra returned to the absorbance spectrum of the blank medium. A nanomaterial is defined as a material where 50% or more of the particles—in an unbound state, as an aggregate or as an agglomerate—have one or more external dimensions in the size range 1–100 nm [46,47]. TEM showed aggregates with sizes that far exceed the limits of this definition. Moreover, DLS results corroborated the instability: high polydispersity was measured, and size distribution graphs were shifting and were situated finally on the right side of the size axis. The IDL medium had a neutral pH and contained only salts and glucose. No complex and undefined components were added to this minimal medium. The salts dissociated in ions and this ionic strength led to the compression of the EDL of the electrostatically and electrosterically stabilized AgNPs. The reduction in thickness of this layer, causing a decrease in repulsive electrostatic force and, thus, aggregation can occur more easily [30,31,48,49,50]. Moreover, both negatively and positively charged ions present in the medium interacted with charged groups of the stabilizing agents. Citrate groups are negatively charged, for example, and the electrosteric stabilizing agent BPEI has a positive charge. Via this interaction, the surface of the AgNPs is neutralized, the repulsive forces are weaker and, thus, aggregation can occur [48,51,52]. Moreover, previously reported studies revealed that the behavior of the PVP polymer depends on the ionic composition, and ions like H_2_PO_4_^−^, SO_4_^2−^ and HPO_4_^2−^—highly present in the IDL medium—can have a negative influence on the PVP polymer [53,54,55]. This explains, at least partly, why the PVP AgNPs aggregated in the IDL medium. To contrast, previous research [27,56,57,58] indicates that sterically stabilized AgNPs are more stable than other stabilization mechanisms, even at a higher ionic strength. However, this was not confirmed in the IDL, possibly for two reasons. First, the above-mentioned ions that can have a negative influence on the PVP polymer were absent in the referred researches. Moreover, and presumably the most important reason: the IDL medium had a much higher ionic strength than the ones performed in the referred researches. Notwithstanding, the IDL medium can be used as a standard medium for determining Ag^+^ toxicity [20], the NP instability should be taken into account when performing AgNPs toxicity studies within this medium.

Except for two of the tested AgNPs, mixing with LB resulted in aggregation during the performed 24 h. The first exception was observed for the 10 nm NaC AgNPs. When 10 nm NaC AgNPs were suspended in the LB, aggregation occurred immediately but, over time, a disaggregation process was observed through all three techniques. Secondly, the 100 nm BPEI stabilized AgNPs remained stable in the LB medium: no color change or change in spectrum was observed when time advanced and TEM and DLS data showed no aggregation. NaC stabilized AgNPs cause a different stability depending on their size. Aggregation of the 100 nm NaC AgNPs possibly occurred due to the disturbance of the EDL [30,31,48,49,50], as previously described. Ten nanometers of NaC AgNPs behaved differently and disaggregated when time advanced. Compared with IDL, the LB has a similar and neutral pH, but a higher EC value, and contains organic matter including hydrolyzed casein and yeast extract, both high in protein [59,60,61]. It is assumed that the original electrostatically NaC stabilized particles first aggregate due to the EDL compression followed by the adsorption of organic molecules onto the surface of the AgNPs, resulting in a steric restabilization. A similar result was reported by Albanese et al. for gold NPs [50]. We assume that the higher surface area of the 10 nm AgNPs compared with the 100 nm AgNPs leads to more interaction with the organic matter and, thus, a size-dependent aggregation response. Analogous to IDL, the ionic composition (in this case NaCl) and ionic strength of the LB medium led to an instability of both sizes of the PVP AgNPs [53,54,55]. However, it was remarkable that both 10 and 100 nm PVP AgNPs initially showed no aggregation immediately after mixing with LB but, as soon as the incubation time increased, the aggregation occurred. A better stability was thus initially achieved when organic matter was present, even at a high ionic strength. The enhanced NP stability due to an additional sterical hindrance by the presence of organic compounds already was reported by other researchers [62,63,64,65,66]. Similar to the NaC AgNPs, BPEI stabilized AgNPs showed a size dependent stability. Different sizes lead to a different curvature of spherically shaped AgNPs and to a difference in the physical packing of the electrosterically stabilizing agent. Smaller sized AgNPs have a higher curvature compared to larger sized AgNPs. This higher curvature leads to a reduced layer thickness of the polyelectrolytes (like BPEI) and fewer interaction places of stabilizer—NP surface [67,68,69]. Hence, lower stability can be observed for smaller AgNPs. This phenomenon was confirmed by the BPEI AgNPs: the LB medium led to the disturbance of the BPEI stabilizing mechanism for 50 nm AgNPs, while the 100 nm AgNPs remained stable through time.

## 5. Conclusions

We provided a case study here, where the stability of AgNPs with different stabilization mechanisms and sizes was analyzed within bacterial growth media during 24 h. It has become clear that the effect of (1) aging, (2) medium composition (the environment), (3) the NP size and (4) the NP stabilization mechanism has a profound influence on the stability. Our results showed that the addition of complex organic matter to the environment led to a better stability for some of the tested AgNPs—even at a high ionic strength—possibly due to an extra sterical NP hindrance [62,63,64,65,66]. Moreover, we proved that the stability was size-dependent, attributed to the difference in curvature or surface area of small versus larger AgNPs [2,18,50,67,68,69,70]. Finally, the stabilization mechanism of the AgNPs was important. Different stabilized AgNPs behave differently and the aggregation was dependent from the incubation time. To the best of our knowledge, this is the first report to demonstrate the influence of all 4 parameters on the stability of AgNPs.

The impressive difference in toxicity of MNPs within different environments already was noted by some other researches [31,50,71,72]. We believe that the difference in MNP stability within these different environments is at least partly responsible for the observed toxicity differences. NPs are characterized by a large surface-to-volume ratio due to their smaller size-order in comparison to the bulk material [2,18,70]. When NPs aggregate, the external and reactive surface area is decreasing and, therefore, the reactivity, bioavailability and toxicity changes [4,8,31,50]. MNP stability, therefore, should be considered within MNPs research. The MNP stability should be analyzed within their application during storage, on the one hand. Conversely, and at least as important, is analyzing the MNP stability in the environment that is achieved when the MNPs move from their application to their intentional or unintentional ‘target’. The complexity of these ‘target’ environments with a certain pH, ionic strength and often a presence of organic matter is extremely diverse and can range from ground water, for example, to active sludge [3,29,73,74] or from human gastrointestinal fluids to blood [12,15].

We believe that our approach was much needed because the influence of (1) aging, (2) medium composition (the environment), (3) the NP size and (4) the NP stabilization mechanism on the MNPs stability currently is underrepresented in literature [1,4,15,26,27,28]. Since we proved that all four of the aforementioned parameters have a high relevance on MNPs stability, we strongly recommend their inclusion in future projects. Authors should monitor the MNP (with a certain size and stabilization mechanism) stability within the specific application domain or simulated circumstances during the performed time period and take these results into account before making conclusions. 

## Figures and Tables

**Figure 1 nanomaterials-09-01684-f001:**
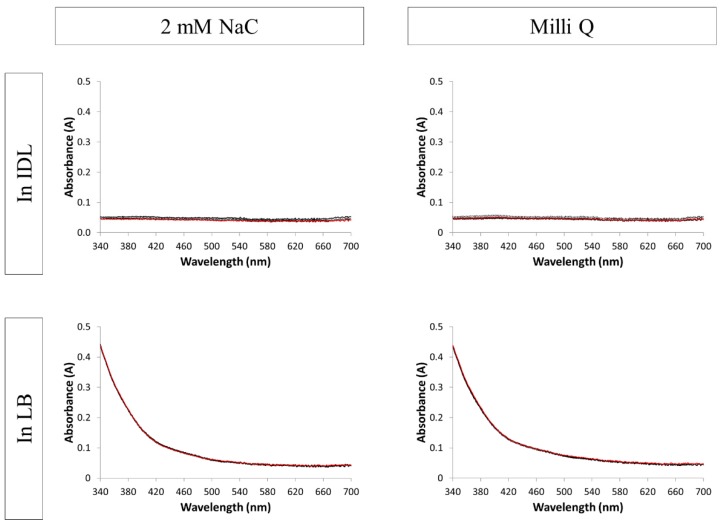
Absorption spectra of the blanks (without silver nanoparticles). Overlays of different time points between 0 h and 24 h are shown. The first (black) and last (red) measurement is represented by a solid line. (IDL = minimally defined medium; LB = Luria-Bertani, complex undefine medium; NaC = sodium citrate).

**Figure 2 nanomaterials-09-01684-f002:**
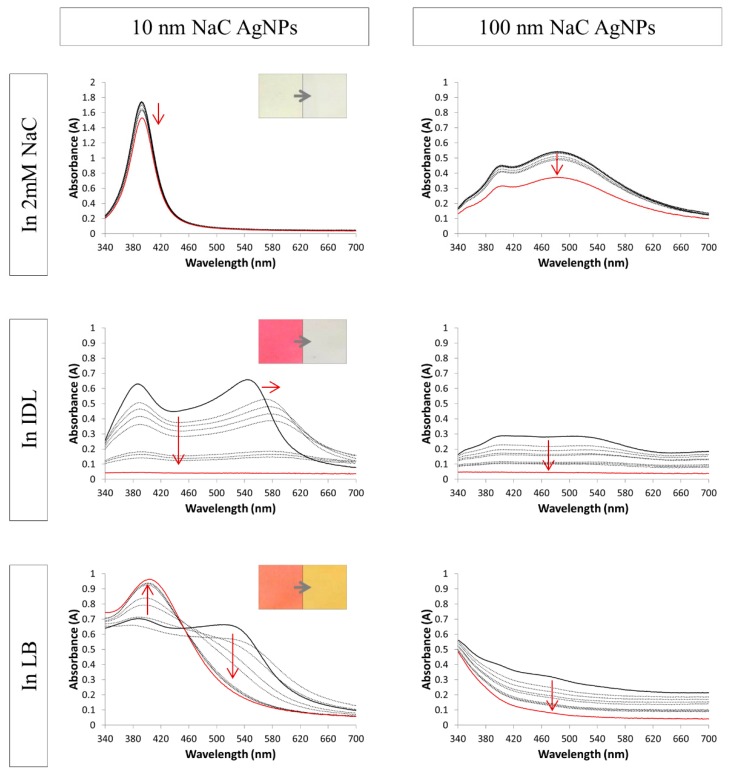
Absorption spectra of 10 nm (**left**) and 100 nm (**right**) NaC (sodium citrate) stabilized AgNPs (silver nanoparticles) in 2 mM NaC, IDL (minimally defined medium) and LB (Luria-Bertani, complex undefined medium). Overlays of different time points between 0 h and 24 h are shown. The first (black) and last (red) measurement is represented with a solid line, the arrows indicate the shift. The inserts show the color of the 0 h and 24 h timepoint.

**Figure 3 nanomaterials-09-01684-f003:**
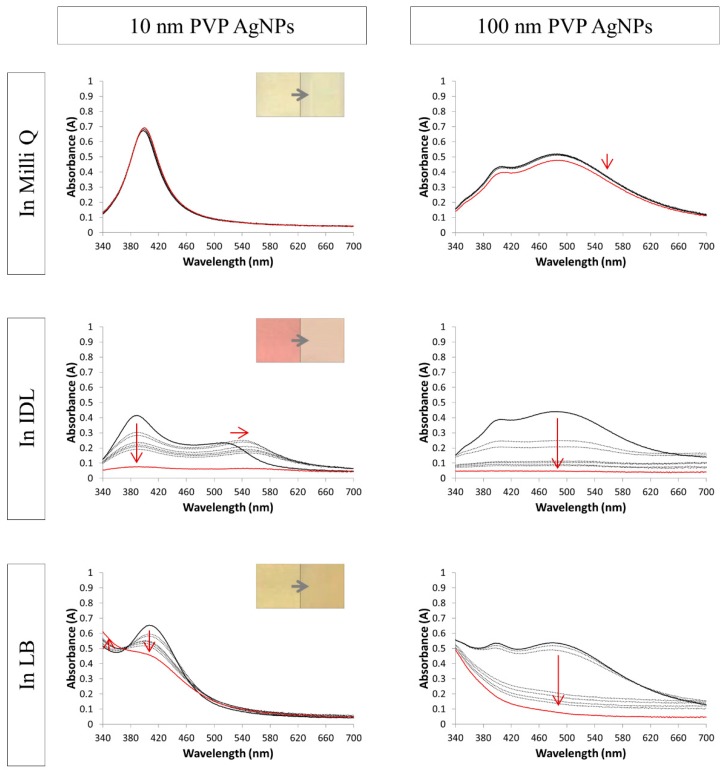
Absorption spectra of 10 nm (**left**) and 100 nm (**right**) PVP (polyvinylpyrrolidone) stabilized AgNPs (silver nanoparticles) in Milli Q, IDL (minimally defined medium) and LB (Luria-Bertani, complex undefined medium). Overlays of different time points between 0 h and 24 h are shown. The first (black) and last (red) measurement is represented with a solid line, the arrows indicate the shift. The inserts show the color of the 0 h and 24 h timepoint.

**Figure 4 nanomaterials-09-01684-f004:**
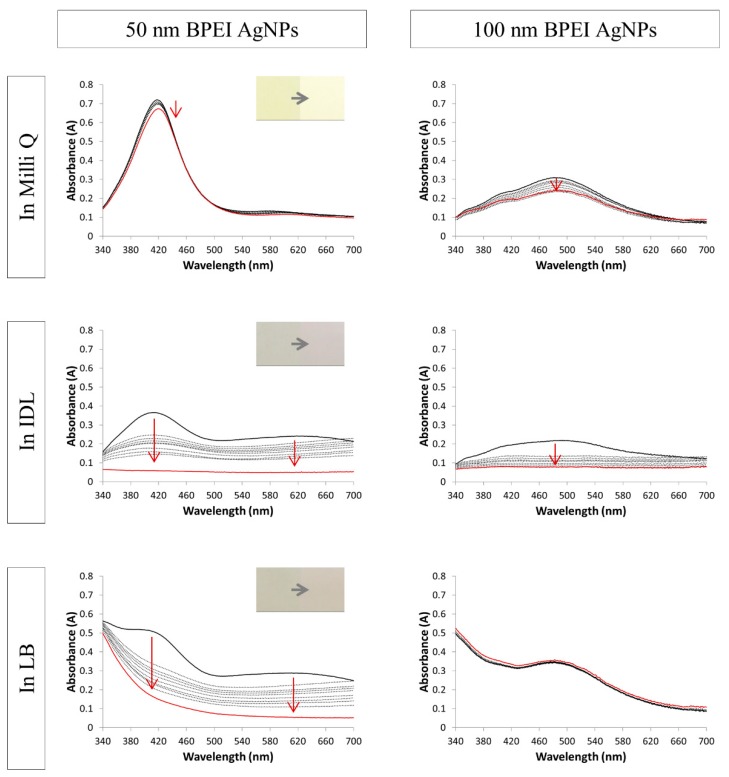
Absorption spectra of 50 nm (**left**) and 100 nm (**right**) BPEI (branched polyethyleneimine) stabilized AgNPs (silver nanoparticles) in Milli Q, IDL (minimally defined medium) and LB (Luria-Bertani, complex undefined medium). Overlays of different time point between 0 h and 24 h are shown. The first (black) and last (red) measurement is represented with a solid line, the arrows indicate the shift. The inserts show the color of the 0 h and 24 h timepoint.

**Figure 5 nanomaterials-09-01684-f005:**
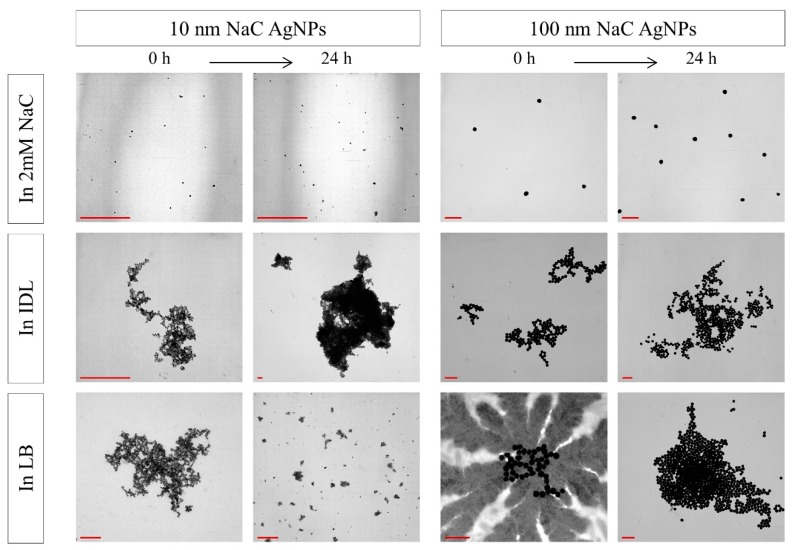
TEM images of 10 nm (**left**) and 100 nm (**right**) NaC (sodium citrate) stabilized AgNPs (silver nanoparticles) in 2 mM NaC, IDL (minimally defined medium) and LB (Luria-Bertani, complex undefined medium), immediately or after 24 h of incubation. The scale bar represents 400 nm.

**Figure 6 nanomaterials-09-01684-f006:**
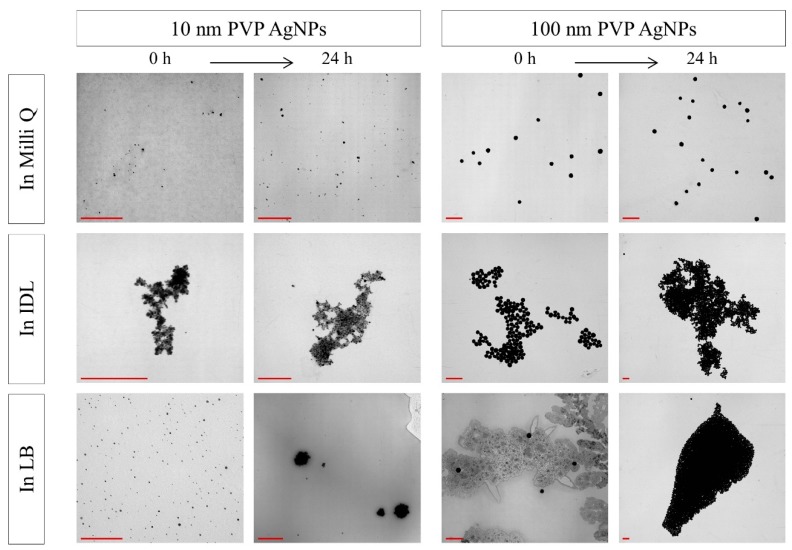
TEM images of 10 nm (**left**) and 100 nm (**right**) PVP (polyvinylpyrrolidone) stabilized AgNPs (silver nanoparticles) in Milli Q, IDL (minimally defined medium) and LB (Luria-Bertani, complex undefined medium), immediately or after 24 h of incubation. The scale bar represents 400 nm.

**Figure 7 nanomaterials-09-01684-f007:**
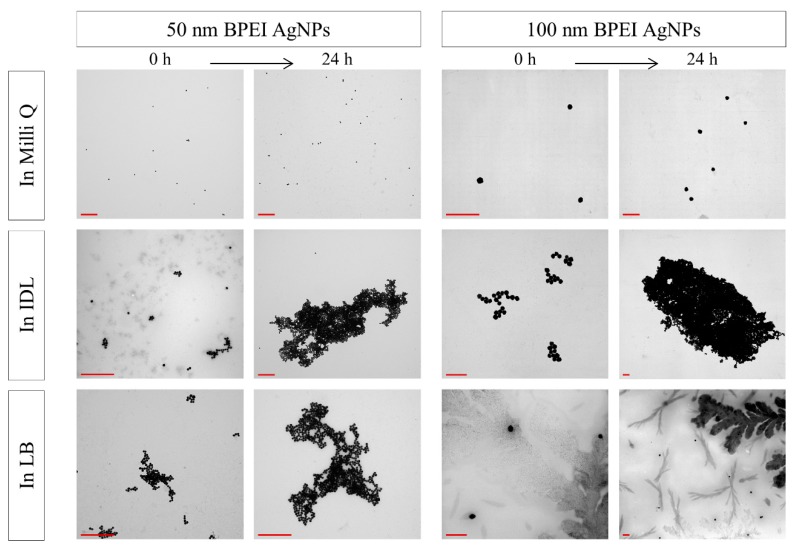
TEM images of 50 nm (**left**) and 100 nm (**right**) BPEI (branched polyethyleneimine) stabilized AgNPs (silver nanoparticles) in Milli Q, IDL (minimally defined medium) and LB (Luria-Bertani, complex undefined medium), immediately or after 24 h of incubation. The scale bar represents 400 nm.

**Figure 8 nanomaterials-09-01684-f008:**
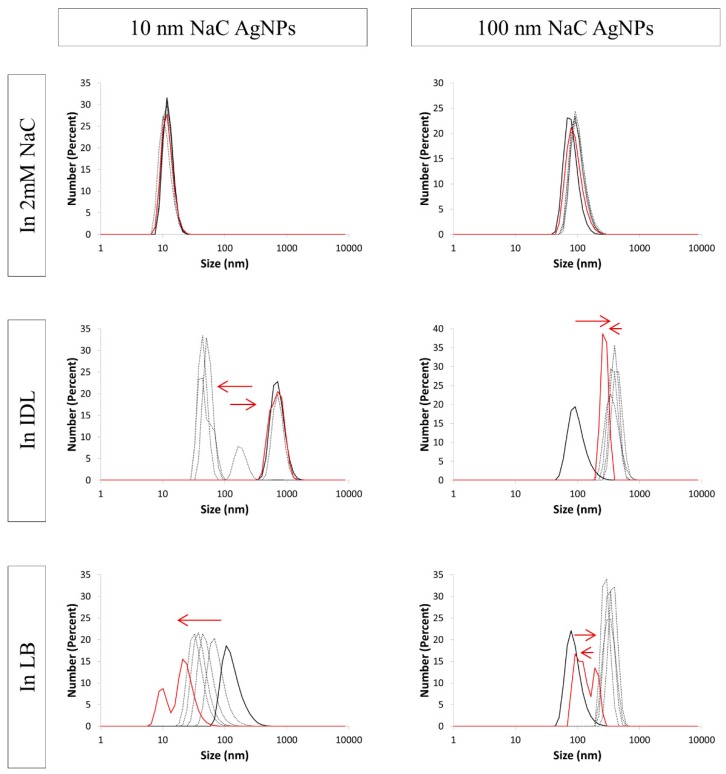
Number distribution of 10 nm (**left**) and 100 nm (**right**) NaC (sodium citrate) stabilized AgNPs (silver nanoparticles) in 2 mM NaC, IDL (minimally defined medium) and LB (Luria-Bertani, complex undefined medium). Overlays of different time points between 0 h and 24 h are shown. The first (black) and last (red) measurement is represented with a solid line, the arrows indicate the shift.

**Figure 9 nanomaterials-09-01684-f009:**
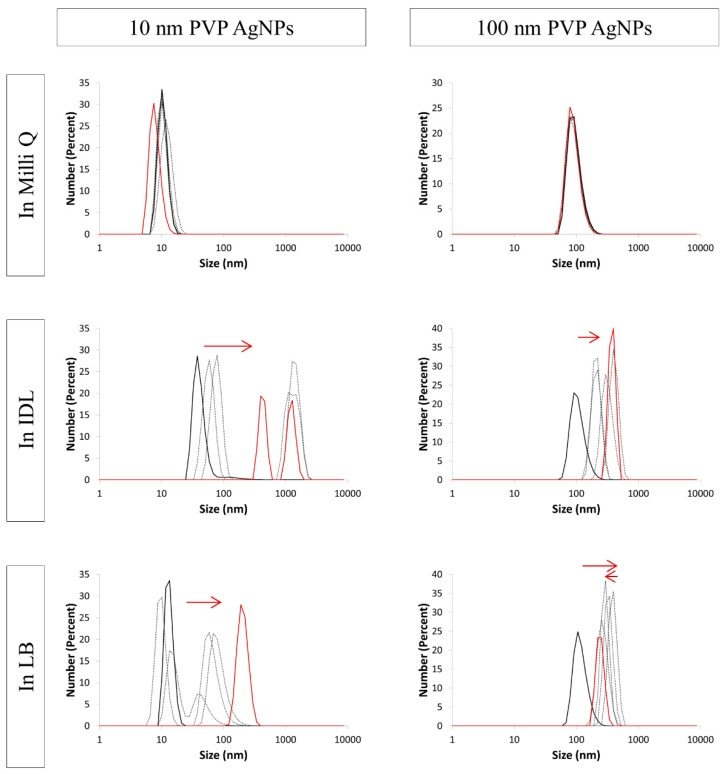
Number distribution of 10 nm (**left**) and 100 nm (**right**) PVP (polyvinylpyrrolidone) stabilized AgNPs (silver nanoparticles) in Milli Q, IDL (minimally defined medium) and LB (Luria-Bertani, complex undefined medium). Overlays of different time points between 0 h and 24 h are shown. The first (black) and last (red) measurement is represented with a solid line, the arrows indicate the shift.

**Figure 10 nanomaterials-09-01684-f010:**
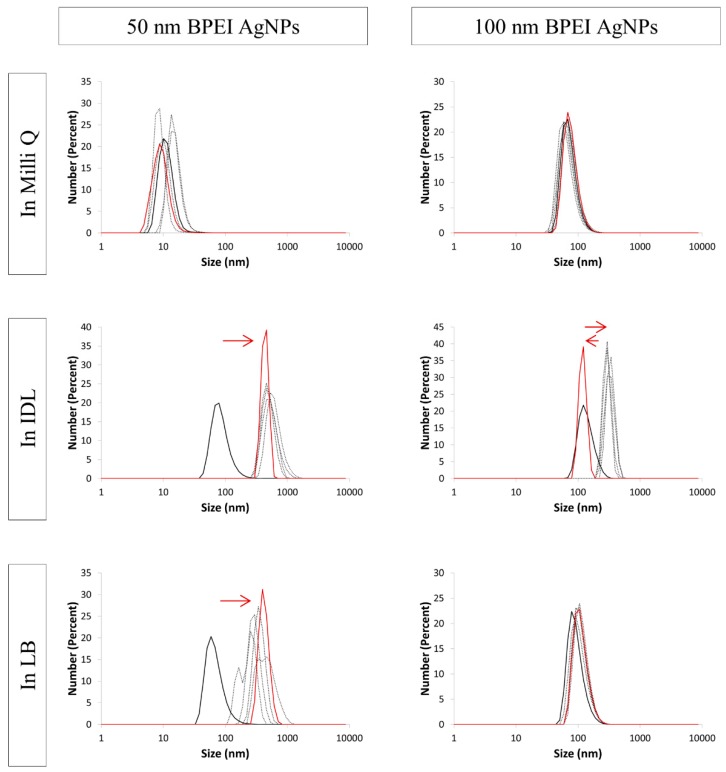
Number distribution of 50 nm (**left**) and 100 nm (**right**) BPEI (branched polyethyleneimine) stabilized AgNPs (silver nanoparticles) in Milli Q, IDL (minimally defined medium) and LB (Luria-Bertani, complex undefined medium). Overlays of different time points between 0 h and 24 h are shown. The first (black) and last (red) measurement is represented with a solid line, the arrows indicate the shift.

**Table 1 nanomaterials-09-01684-t001:** PdI (Polydispersity Index) of NaC (sodium citrate) (10 and 100 nm), PVP (polyvinylpyrrolidone) (10 and 100 nm) and BPEI (branched polyethyleneimine) (50 and 100 nm) stabilized AgNPs (silver nanoparticles) in their solvent, IDL (minimally defined medium) and LB (Luria-Bertani, complex undefined medium). PdI was measured at different time points between 0 h and 24 h. PdI values > 0.400 are indicated by grayscale.

	**NaC AgNPs**
	**10 nm**	**100 nm**
	**In 2 mM NaC**	**In IDL**	**In LB**	**In 2 mM NaC**	**In IDL**	**In LB**
0 h	0.371	0.340	0.172	0.055	0.191	0.189
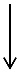	0.316	0.686	0.155	0.067	0.379	0.564
0.312	0.552	0.174	0.077	0.645	0.648
0.289	0.510	0.191	0.078	0.782	0.797
0.423	0.698	0.215	0.056	0.700	0.965
24 h	0.200	0.948	0.233	0.119	1.000	1.000
	**PVP AgNPs**
	**10 nm**	**100 nm**
	**In Milli Q**	**In IDL**	**In LB**	**In Milli Q**	**In IDL**	**In LB**
0 h	0.200	0.327	0.240	0.058	0.076	0.029
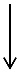	0.205	0.693	0.188	0.048	0.461	0.449
0.195	0.757	0.319	0.071	0.669	0.682
0.196	0.575	0.245	0.065	0.780	0.704
0.168	0.620	0.213	0.046	0.882	0.906
24 h	0.384	0.844	0.487	0.050	1.000	1.000
	**BPEI AgNPs**
	**50 nm**	**100 nm**
	**In Milli Q**	**In IDL**	**In LB**	**In Milli Q**	**In IDL**	**In LB**
0 h	0.324	0.217	0.226	0.105	0.262	0.084
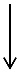	0.352	0.460	0.459	0.126	0.578	0.107
0.335	0.504	0.599	0.172	0.946	0.090
0.360	0.856	0.795	0.156	0.976	0.035
0.333	0.627	0.656	0.174	1.000	0.010
24 h	0.319	1.000	0.917	0.215	1.000	0.092

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
