# Peer review of "Revealing the Importance of Aging, Environment, Size and Stabilization Mechanisms on the Stability of Metal Nanoparticles: A Case Study for Silver Nanoparticles in a Minimally Defined and Complex Undefined Bacterial Growth Medium"

_nanomaterials, 2019, doi:10.3390/nano9121684_

Round 1

Reviewer 1 Report

The article explores very interesting area of role of nanoparticles in science. The authors demonstrate problems with nanoparticles, which can change activity after aggregation. However I have some  remarks for this text.

Firstly, in paragraph Materials and Methods the authors used abbreviations without providing the full name of substance, for example NaC. Secondly, they give no explanation what  the difference is between M9 and IDL mediums. What’s more in lines 100-102 the wrong formulas were used - for example  MgSO4.7H2O, but should be  MgSO4 * 7H2O.

Small  150 Ag nanospheres (10 - 50 nm) typically have a small absorbance peak near 400 nm, while larger spheres  151 (100 nm) give a broader peak with a maximum that shift toward longer wavelengths near 500 nm -  this information, especially absorbance peak for nanospheres, requires citing a source in literature.

References 36-40 are not scientific literature references, but are information about chemical products. Especially, it is surprising when a link to the nutrient components is shown as a literature list – Ref 56-57. For item Ref 42, authors should complete the article information.
In verse 277 is included additional information.

Figure 2 presents data that, in my opinion, should be explained - how do the authors explain the clearly visible peak at 400 nm in the spectrum for 100nm nanoparticles (in 2 mM NaC)? Does this indicate the presence of 10 nm particles in suspension? Similar doubts are caused by the changes shown for the spectrum of 10nm particles suspended in LB - how to explain the increase in absorbance at 400nm (peak characteristic for 10nm particles)? It is all the more interesting that the remaining graphs show a decrease in absorbance at 400 nm.
I would like the authors' comment.
Similar doubts arise in the case of the graphs shown in Figure 8. And 9. Do 10nm NaC AgNPs or PVP AgNPs disintegrate when placed in LB medium a?
Are the observed changes for 100nm AgNPs in LB medium related to the adhesion of medium components to individual nanoparticles?

The presented research combine some analysis techniques  of nanoparticle and it’s a very interesting way, especially they present problems related to their aggregation. This is particularly important from the point of view of the application of such elements and changes in the level of biological activity.
The work prepared by the authors requires several corrections and additional comments on the obtained experimental results.

Reviewer 2 Report

The manuscript seems to be ready to be published.

Author Response

Thank you for reviewing our manuscript entitled ‘Revealing the importance of aging, environment, size and stabilization mechanism on the stability of metal nanoparticles: a case study for silver nanoparticles in a minimal defined and complex undefined bacterial growth medium’. We appreciate your time and are pleased that you had no comments on our submitted manuscript.

Reviewer 3 Report

The article reports a thorough study of the colloidal stability/aggregation of silver nanoparticles in biological media. The authors studied the colloidal stability by three different techniques, UV/vis spectroscopy, TEM and DLS and the results obtained soundly prove the conclusions of the article. Overall, I believe that this work is of interest to many scientists in the areas of nanotoxicology and nanobiotechnology.

Author Response

(The authors gave the same response as above.)
